# Integrative Analysis of Multi-Omics Data Based on Blockwise Sparse Principal Components

**DOI:** 10.3390/ijms21218202

**Published:** 2020-11-02

**Authors:** Mira Park, Doyoen Kim, Kwanyoung Moon, Taesung Park

**Affiliations:** 1Department of Preventive Medicine, Eulji University, Daejeon 34824, Korea; mira@eulji.ac.kr; 2Department of Statistics, Korea University, Seoul 02841, Korea; dydykim@korea.ac.kr (D.K.); kikironi@korea.ac.kr (K.M.); 3Department of Statistics, Seoul National University, Seoul 08826, Korea

**Keywords:** dimensional reduction, multi-omics data, sparse principal component analysis, variable clustering

## Abstract

The recent development of high-throughput technology has allowed us to accumulate vast amounts of multi-omics data. Because even single omics data have a large number of variables, integrated analysis of multi-omics data suffers from problems such as computational instability and variable redundancy. Most multi-omics data analyses apply single supervised analysis, repeatedly, for dimensional reduction and variable selection. However, these approaches cannot avoid the problems of redundancy and collinearity of variables. In this study, we propose a novel approach using blockwise component analysis. This would solve the limitations of current methods by applying variable clustering and sparse principal component (sPC) analysis. Our approach consists of two stages. The first stage identifies homogeneous variable blocks, and then extracts sPCs, for each omics dataset. The second stage merges sPCs from each omics dataset, and then constructs a prediction model. We also propose a graphical method showing the results of sparse PCA and model fitting, simultaneously. We applied the proposed methodology to glioblastoma multiforme data from The Cancer Genome Atlas. The comparison with other existing approaches showed that our proposed methodology is more easily interpretable than other approaches, and has comparable predictive power, with a much smaller number of variables.

## 1. Introduction

Recent advances in high-throughput technologies have generated massive amounts of various types of biological data. Moreover, multiple “omics” data, such as genomics, epigenomics, transcriptomics, proteomics, and metabolomics, have been collected from various sources [1,2,3]. Analysis of these omics data sets may provide insight into mechanisms of disease, or the identification of new biomarkers [4]. However, because omics data is high-dimensional, there are many difficulties in using all the variables, to establish a statistical model for diagnostic or prognostic purposes. For example, computational instability, variable redundancy, and difficulties in interpretation can all occur. Moreover, integration of multiple omics data is also a challenge. In this paper, we propose an approach for constructing an efficient prediction model, for multi-omics data, considering three aspects: dimensionality reduction, variable selection, and data integration.

Dimension reduction plays a crucial role in data exploration, downstream pattern recognition, classification, and clustering for high-dimensional data that contains a large number of variables [4,5,6,7]. With regard to unsupervised approaches, principal component analysis (PCA) is a well-known dimensional reduction method, providing principal components (PCs) that are derived by linearly combining the original variables. The coefficients of the original variables are called “loadings” [8]. Instead of using individual variables, several PCs can be used as predictors for disease modeling.

However, each PC still needs all the original variables, since the coefficients are typically nonzero [9], and it is difficult to interpret the derived PCs. Therefore, it is desirable to select variables, as well as to reduce the dimensionality. An ad hoc approach is to ignore variables with small absolute values of loadings, when interpreting the PCs, or to artificially set the loading of variables, that are less than a certain threshold, to zero [10]. Though frequently used in practice, this approach is quite unreliable, since it does not take into account the variances, and the patterns of correlations of the variables [11]. The second possible approach is to use regularization penalties, to modify principal components, with sparse loading. Sparse principal component analysis (sPCA), which imposes elastic net penalties, can be used to overcome the drawback of PCA, and be formulated for variable selection [10]. To apply sPCA for ultra-high dimensional data, such as omics data, a screening procedure is still required to perform a penalized selection. One approach is to filter significant variables first, by repeating single supervised analysis. However, this type of variable selection can repeatedly select highly correlated variables, although the problems of variable redundancy and collinearity between variables, remain [12,13].

On the other hand, it is common to analyze single omics data sets, from a single study; however, to better understand the mechanisms for complex diseases, integration of the same types of omics data, across multiple studies, and/or integration of multi-omics data for the same samples, is desirable [2,14,15,16]. For horizontal integration of the same type of omics data, typical meta-analysis can be used for biomarker detection [17,18]. For vertical integration of multi-omics data, parallel integration, which treats each type of omics measurements equivalently, has been studied [19]. Usually, more attention is focused on vertical integration than horizontal integration. As for supervised integration, Zhao et al. [20] integrated four types of multi-omics datasets from The Cancer Genome Atlas (TCGA), including gene mRNA expression, DNA methylation, microRNA expression, and copy number alterations, for predicting cancer prognosis of invasive breast carcinoma, glioblastoma multiforme, acute myeloid leukemia, and lung squamous cell carcinoma. They then extracted important features by applying PCA, partial least squares (PLS), and least absolute shrinkage and selection operator (LASSO) methods, for individual genomic data sets, and then fitted them to a Cox model [20]. For the prediction of cutaneous melanoma prognosis, Jiang et al. [21] used elastic net, sparse PCA, and sparse PLS, to extract important variables from each type of omics dataset, and then integrated the important variables. Please note that in both studies, supervised screening for single genes was conducted, for each omics data set, in advance, to select a small number of genes. Therefore, they did not avoid the problems of redundancy and collinearity, of selected variables.

Various methodologies have been developed for dimensional reduction, variable selection, and integration of multi-omics data. Before conducting an integrative analysis of multi-omics data, variable selection in each single omics dataset is usually done first, whether it is explicitly reported or not [19]. Supervised techniques involving penalized variable selection such as LASSO, SCAD, and MCP can be applied [22,23,24,25]. Methods based on Bayesian approaches can also be applied for biological data [26]. In addition, some machine learning techniques, including the random forest and boosting methods, are applicable for variable selection in integrative analysis [27,28]. Unsupervised techniques, such as PCA and canonical correlation analysis, have also been considered. Several variations of PCA, such as consensus PCA (CPCA), multiple-block PCA (MBPCA), and nonnegative matrix factorization (NMF) have been developed [29,30,31]. By adding sparsity properties to these methods, sPCA, sparse CCA, and the joint and individual variation explained (JIVE) method were applied for genomic data [32,33,34]. There are various tools and methodologies for multi-omics data integration [16,19,35]. Wu et al. [19] classified integrated analyses into parallel and hierarchical integration. Parallel integration treats each omics dataset equally, whereas hierarchical integration incorporates the prior knowledge of regulatory relationships among each omics dataset. For example, iCluster, an integrative clustering method using a joint latent variable model, and its variants exemplify the parallel integration approach [36], while iBAG, an integrative Bayesian analysis of genomics data, is a hierarchical integration method [37]. Subramanian et al. [38] recently introduced a total of 31 analysis tools classified into similarity, correlation, network, Bayesian, fusion and multivariate approaches. Although many methodologies have been developed, each has its own advantages and disadvantages, and none of them dominates the others.

In this study, we propose an integrative analysis of multi-omics data that reduces multi-collinearity and redundancy, while reducing dimensionality and selecting variables of importance. It is based on blockwise sparse principal components. The proposed methodology is called iMO-BSPC (Integrative analysis of Multi-Omics data based on Block-wise sparse Principal Component Analysis). Unlike other multi-omics integration methods, our approach first performs variable clustering for each omics data. This process yields a set of closely related variables, and provides the same information contained in the initial variables. Herein, we call this a variable cluster “block”. For each block, we select the first sparse principal components (sPCs), as surrogate variables for dimensional reduction. Applying sPCA, instead of PCA, can reduce the number of variables. Then, sPCs from each omics data are merged to form an integrated multi-omics dataset. Finally, a prediction model using these sPCs as explanatory variables is constructed. We also suggest a new graphical method based on polar charts, that represents the results from sPCA and Cox regression. The proposed graph can simultaneously represent important sPCs for survival time prediction and the variables that make up the sPCs.

In Section 2, we apply the iMO-BSPC to glioblastoma multiforme (GBM) data sets from the TCGA. Since the response variable is survival time, with censoring, we perform Cox regression analysis for prognosis. Comparisons of the results from two alternative approaches for dimensional reduction are considered. We summarize and discuss the results in Section 3. The step-by-step procedure of the proposed iMO-BSPC approach is introduced in Section 4.

## 2. Results

We applied our framework to glioblastoma multiforme (GBM) data from The Cancer Genome Atlas (TCGA). The data set consists of 215 GBM patients, and three omics data: DNA methylation, with 1305 genes, mRNA expression with 12,042 genes, and miRNA expression, with 534 miRNAs. Among the 12,042 genes’ mRNA expression data, we sorted the genes in alphabetical order and selected the first gene for every 10 genes, i.e., 1205 genes in total. As a response variable, we used survival times, and their corresponding censoring statuses. We then used data downloaded from http://compbio.cs.toronto.edu/SNF/SNF/Software.html.

### 2.1. Analysis Using iMO-BSPC

We first conducted variable clustering, using the hclustvar() function in the R package ‘ClustOfVar’(ver. 1.1). Based on the dendrograms and scree plots, of each omics data set, we determined 12, 10, and 10 blocks for DNA methylation, mRNA expression, and miRNA expression, respectively. Figure 1 shows the dendrograms and scree plots for each omics data set. In dendrograms, each block is depicted by a rectangle. As a result of variable clustering, we obtained 32 homogeneous variable blocks.

Next, sPCA was performed on each block. The first sPC was obtained from 32 blocks, resulting in a total of 32 sPCs. We used 10% of the number of variables as sparsity threshold. Thus, each sPC was made up of 10% of its original variables. This procedure was performed using the spca() function in the R package ‘elasticnet’. Appendix A shows the obtained sparse loadings. For convenience, only the results corresponding to the block selected in the next stage are listed. For example, the sPC in the fourth block, DNA methylation (D4), consisted of five genes of ALK_E183_R, KRAS_P651_F, IFNGR2_E164_F, RASA1_E107_F, and CCKBR_P361_R. Among them, ALK_E183_R showed the highest influence from sparse principal components, among the variables in the block. KRAS_P651_F, the variable with the second largest loading, represented the variable of DNA methylation block 4. All the selected variables from each block are listed in Appendix A.

For the integrated dataset with 32 sPCs for 215 patients we used the Cox regression model. Stepwise variable selection based on Akaike information criterion (AIC) values was performed. Thirteen sPCs were finally selected that had significant effects on survival time. The AIC was 1731.66. Table 1 summarizes the selected sPCs, and their coefficients and hazard ratios (HRs). SPC of mRNA’s 9th block (MR9) showed the biggest absolute coefficient among 34 blocks, followed by the 2nd block of DNA methylation (D6), the 3rd block of miRNA (MI3), and so on. For example, sPC of mRNA’s 9th block (MR9) had a hazard ratio of 0.67, showing a significant effect on survival time. The larger the value of sPC of MR9, the lower the HR.

After fitting the Cox regression model, we drew a multi-level polar chart (MP chart), and identified biomarkers related to GBM. Figure 2 shows the MP chart of the results from the integrated dataset. Using this graph, we can easily assess the effects of sPCs on survival time. The illustrative plot depicts that the sPC of the 9th block of mRNA (MR9) had an outstanding effect on survival, followed by sPC of the 6th block of DNA (D6), the 3rd block of miRNA (MI3), and so on. It also shows that the sPCs of MR9 are mainly composed of LAIR1 and MNDA. The coefficients of the two genes, also being negative, shows that the larger the value of these genes, the smaller the value of the sPC. On the other hand, the sign of the sPC for the MR9 coefficient was negative, meaning that the HR decreases, as the value of the sPC for MR9 increases. Therefore, we can interpret that when the values of LAIR1 and MNDA are large, HRs tend to increase and survival times tend to decrease. SPC of D6 was composed of various variables, including BCL3_E71_F. Please note that hsa-miR-602 and hsa-miR-17-5p dominated the sPCs for the 3rd and 7th blocks of miRNA data, respectively.

Using the Cox regression model, we predicted whether any specific GBM patient’s survival time is above or below the median survival time. To that end, we assessed the predictive power of the proposed methodology, using a cumulative/dynamic ROC curve and its corresponding AUC value. The cumulative/dynamic ROC curve derived from Cox regression of integrated data is shown in Figure 3. The corresponding AUC value was 0.74 and C-index was 0.67.

### 2.2. Comparison with Other Approaches

We compared the predictive power of our framework with other methodologies. We focused on comparing our method with others using similar approaches. In particular, we considered (i) PCA analyses without variable clustering, (ii) traditional PCA with clustering, and (iii) iMO-BSPC for each set of omics data. For the first approach, we performed traditional PCA for each omics dataset. That is, neither variable clustering to form homogeneous blocks nor shrinkage to reduce the number of variables was performed. Then, we conducted a Cox regression model with PCs. For the second approach, after variable clustering, we performed PCA for each block. Here, the difference lies in applying PCA instead of sPCA. Finally, in order to compare the results from multi-omics dataset analysis against those from single omics datasets, we applied iMO-BSPC to each omics dataset. For these existing methods, we constructed a Cox model (a) for each omics dataset, and (b) for the entire multi-omics dataset. Therefore, a total of 6 (= 3 × 2) methods, including the proposed method, were compared. Table 2 summarizes the schemes and results of the comparison.

Single omics dataset analysis was also performed, for the purpose of comparison. Since the iMO-BSPC methodology created variable blocks for each omics dataset, there were 12, 10, and 10 blocks for DNA methylation, mRNA expression, and miRNA expression, respectively. During the variable selection process, four sPCs from DNA methylation, four sPCs from mRNA expression, and three sPCs from miRNA expression were finally chosen for building a prediction model. The MP chart generated by iMO-BSPC, from three single omics dataset analyses, is depicted in Figure 4. For DNA data, the sPC of D5 had the greatest influence, followed by the sPCs of D6, D2, and D11. Compared to the multi-omics results, the sPCs of D4 and D9 were removed, while the sPC of D2 was added for single omics. D5 and D6 had coefficients of similar magnitude, but the signs were opposite. For mRNA, sPC of M2 had the greatest influence, followed by the sPCs of M3, M6, and M5. Single omics dataset analysis of mRNA chose sPCs that were all different from those chosen in the multi-omics data analysis. In contrast, for the miRNA datasets, the same sPCs were chosen, except for MI10. Here, the effect of MI7 was shown to be larger than that of the multi-omics case. The AUCs for the DNA, mRNA, and miRNA datasets were 0.66, 0.60, and 0.63, respectively, much lower than the AUC value of 0.74 from the multi-omics data analysis. The C-indices were 0.61, 0.59, and 0.60 for the DNA methylation, mRNA, and miRNA datasets, respectively (Table 2).

When applying traditional PCA without variable clustering, 188 PCs had an eigenvalue of 1 or higher, and the first 78 PCs accounted for more than 80% of the total variance for DNA methylation. For mRNA expression and miRNA expression, 120 and 77 PCs had eigenvalues of 1 or higher, and the first 32 and 39 PCs explained more than 80% of total variance, respectively. For a fair comparison, we used the same number of PCs with the number of sPCs selected by iMO-BSPC. Therefore, the first 12, 10, and 10 PCs were used to build the prediction model. During the variable selection process, 5, 2, and 2 PCs remained for the DNA methylation, mRNA expression, and miRNA expression dataset, respectively. The final PCs in the prediction model were linear combinations of 563, 188, and 225 variables for each omics dataset, respectively, and the number of variables used was much more than that of iMO-BSPC, as seen in Table 2. However, the AUCs and C-indices were rather lower than those of iMO-BSPC, except for the DNA methylation data.

When applying traditional PCA to the blocks after variable clustering, total of 15 PCs with 1661 variables were selected for the prediction model. Compared to iMO-BSPC, the AUC and C-index values were higher (AUC = 0.71, C-index = 0.64) for DNA methylation, from which twice as many PCs were extracted, but the AUC and C-index values were lower than those of iMO-BSPC for both mRNA and miRNA.

For the multi-omics data analysis, we used the first 32 PCs (sPCs) for each methodology. As seen in Table 2, in all methodologies, the AUC and C-index were higher than the values of the single omics datasets, indicating that it is reasonable to use multi-omics data. The comparison study showed that iMO-BSPC had similar predictive performance (AUC = 0.74, C-index = 0.67) to existing PCA-based methods with a much smaller number of variables.

## 3. Discussion

In this study, we proposed a novel iMO-BSPC (Integrative analysis of Multi-Omics data based on Block-wise sparse Principal Component Analysis) approach for analyzing multi-omics data. Our proposed iMO-BSPC has several advantages. To reduce variable redundancy, iMO-BSPC adopted a shrinkage approach, sPCA, which enabled reductions in dimensionality and the number of explicitly used variables. In addition, the selected variable, the corresponding block, and the results of model fit were expressed in a single graph to enable more intuitive interpretation. The comparison of prediction performance with those of other existing approaches showed that our proposed methodology had comparable AUCs to those of other existing PCA-based methods, with a much smaller number of variables.

However, there are some limitations of our approach. Despite its usefulness in practical applications, sPCA is limited in terms of lack of orthogonality in the loadings of different principal components, the existence of correlations in the principal components, and the expensive computation needed [39]. Also, we used 10% of variables as the threshold. However, more sophisticated methods of choosing the sparsity parameter can be applied. For example, another method chooses the optimal parameter explaining a large proportion of variance with a small number of nonzero loadings among multiple sparsity parameters [10,40]. A sequential method (sPCA-rSVD) using regularized singular value decomposition [41] and an iterative method using penalized matrix decomposition [42] could be used to choose the sparsity parameter. To determine the appropriate number of blocks, we used a simple method of inspecting dendrograms and scree plots. Instead, one may adopt a bootstrap-based approach or use the dissimilarity value or rand index. Another limitation of this study is that it only compared iMO-BSPC with the existing PCA-based method.

iMO-BSPC can be easily extended to various other contexts. If there is a nonlinear relationship, one may apply nonlinear methods, such as kernel PCA, rather than sPCA. Popular kernels, such as Gaussian, polynomial, and hyperbolic tangent kernels can be used [43]. Although we selected Cox regression as a prediction model, our approach can be easily extended to other prediction models and data mining methods such as logistic regression, support vector machine, and random forest. This kind of extension adds high flexibility to our iMO-BSPC and facilitates integrative analyses of multi-omics data.

iMO-BSPC analysis identified several significant genes reported in other studies. The sPCA of 9th clock for mRNA mainly consisted of linear combinations of leukocyte-associated immunoglobulin-like receptor 1 (LAIR1) and human myeloid nuclear differentiation antigen (MNDA). LAIR1 showed the most significant influence among all variables and is known to be broadly expressed in the majority of immune cells [44]. When LAIR-1 binds to its ligands, immune function in the tumor microenvironment is lost, and T cell function and the immune responses of antigen-presenting cells are reduced [45]. Several studies have reported that LAIR1 is significantly upregulated in multiple types of solid tumors such as ovarian cancer, human cervical cancer, and GBM [44,45,46,47,48]. MNDA was the variable with the second largest loading, suggesting that it plays a significant role. MNDA appears to regulate the activity of transcription factors and, in some cases, serves to mediate cell death [49]. Several studies reported that NMDA plays a role as a suppressor of cancer in pancreatic cancer, osteosarcoma, and GBM [50,51,52]. The variables with the largest loadings in DNA methylation cluster 6, DNA cluster 5, miRNA cluster 7, mRNA cluster 10, and miRNA cluster 3 also had significant influences. The representative variable of DNA cluster 6 is BCL3_E71_F. It should be noted that B cell CLL/lymphoma 3 (BCL-3) is a proto-oncogene candidate. BCL3 was reported to be an informative indicator of glioma response to alkylating chemotherapy [53]. It was also identified as a target gene for hepatocellular carcinoma [53,54]. The hsa-miR-602 gene of micro-RNA, homo sapiens miR-602 stem-loop, represented miRNA cluster 3. It has been reported as a high-risk gene for mesenchymal subtype GBM, liver cancer, and hepatocellular carcinoma [55,56,57]. The gene with the largest loading in miRNA cluster 7 was hsa-miR-17–5p, and previous studies have found miR17 to be associated with polycystic kidney disease and B-cell lymphomas. It was identified as a high-risk gene for renal cell carcinoma and colorectal cancer [58,59].

Using 338 pathways in the KEGG database, we performed Fisher’s exact test to identify pathways significantly associated with genes. The Benjamin–Hochberg method was used to adjust the *p*-values. The Fisher’s exact test for the genes in D9 yielded 8 significant pathways with q-values smaller than 5%. Among them, the PI3K-Akt signaling pathway, TNF signaling pathway, and small cell lung cancer have been reported to be related to cancer [60]. In addition, the ECM–receptor interaction, focal adhesion, and cell adhesion molecules pathways were also significant, and could be related to invasive behavior of cancer cells [61]. The Fisher’s exact test for the genes in MR9 yielded Staphylococcus aureus infection as a significant pathway, with a q-value smaller than 5%. It was reported to be related to infection with *S. aureus*.

## 4. Materials and Methods

Our iMO-BSPC approach consists of two stages. In the first stage, we divide whole variables into non-overlapping, homogeneous blocks, using a variable clustering technique, and then construct shrunken components for each block by applying sPCA. This procedure is repeated for each single omics data set. In the second stage, we generate a new data set, by parallel combination of sPCs, from each omics data set. Then, we conduct Cox regression analysis, to construct a prognostic model. The detailed algorithm is given as follows (Figure 5).

### 4.1. Stage 1: Finding Blockwise Components for Each Omics Data

The aim of the first stage is to find a few representative components, for each omics dataset. To accomplish this, we identify homogeneous variable blocks, and then extract sparse PCs.

#### 4.1.1. Finding Homogeneous Variable Blocks

To find homogeneous blocks, we conduct variable clustering, for each omics dataset. There are several methods for clustering, including the VARCLUS procedure of SAS software [62], clustering around latent variables (CLV), and diametrical clustering [42,63]. In this study, we use the ClustOfVar algorithm, which is based on PCAMIX [43,64]. Though it can handle a mixture of qualitative and quantitative variables, we assume that there are only quantitative variables. The algorithm of ClustOfVar packages are as follows.

Consider we have a J omics data set, and that the jth omics dataset $X^{\left(j\right)}$ consists of n samples and $p_{j}$ variables. In our example, we used three omics datasets of DNA methylation, RNA expression and miRNA expression. The DNA methylation data set consisted of 215 samples of GBM patients and 1305 genetic variables. Similarly, the RNA expression and miRNA expression dataset consisted of 1205 and 534 genetic variables for 215 patients, respectively.

Let $P_{k}=\left(C_{1},\cdots ,C_{K_{j}}\right)$ be a partition into $K_{j}$ clusters of $p_{j}$ variables. For each cluster, $C_{k}$, define a synthetic variable $l_{k}$ which is “most linked” to all the variables in the cluster. That is,
(1)$$l_{k}=\underset{u\in R^{n}}{\arg\;\max}\underset{x\in C_{k}}{\sum }r_{x,u}^{2}$$
where $r^{2}\;$denotes the squared Pearson correlation coefficient [64]. The homogeneity of a cluster $C_{k}$ is defined as follows.
(2)$$\;H\left(C_{k}\right)=\underset{x\in C_{k}}{\sum }r_{x,l_{k}}^{2}=\lambda_{1}^{\left(k\right)}$$

Please note that $l_{k}$ is the first principal component of PCAMIX, and $\lambda_{1}^{\left(k\right)}$ is the first eigenvalue obtained, for cluster $C_{k}$ [64]. Now, the aim is to find a partition,$P_{K_{j}}\;$which maximizes the homogeneity function H.
(3)$$\;H=\overset{K_{j}}{\underset{k=1}{\sum }}H\left(C_{k}\right)=\lambda_{1}^{\left(1\right)}+\lambda_{1}^{\left(2\right)}+\cdots +\lambda_{1}^{\left(K_{j}\right)}$$

In this study, we consider the agglomerative hierarchical clustering algorithm, and thus we start with the partition into $p_{j}\;$clusters. Then, we agglomerate two clusters, A and B, with the smallest dissimilarity measure, d, which is defined as:(4)$$\;d\left(A,B\right)=H\left(A\right)+H\left(B\right)-H\left(A\cup B\right)=\lambda_{1}^{A}+\lambda_{1}^{B}-\lambda_{1}^{A\cup B}$$

This aggregation is then repeated until all $p_{j}$ variables belong to a single cluster.

To determine the appropriate number of clusters, we use a simple method for inspecting dendrograms and scree plots. We call the final clusters “blocks” This procedure is conducted, for each omics dataset, to find its homogeneous variable blocks. In the GBM data analysis, we obtained 12 homogeneous gene blocks for DNA methylation dataset, and RNA expression and miRNA expression dataset are divided into 10 homogeneous gene blocks each.

#### 4.1.2. Extracting Sparse Principal Components

To find surrogate variables for each block, we apply sparse principal component analysis (sPCA). SPCA aims to find a set of sparse weight vectors; that is, weight vectors with only a few nonzero values [10,65]. Let $X_{k}^{\left(j\right)}$ be a kth block, of the jth omics data set. Then, the first sparse principal components, $sPC_{1}$, for a data matrix, can be obtained by optimization, as follows:(5)$$\left(\hat{\alpha },\hat{\beta }\right)=\underset{\alpha ,\;\beta }{\arg\;\max}\overset{n}{\underset{i=1}{\sum }}∥x_{i}-\alpha \beta ’x_{i}{∥}^{2}+\lambda ∥\beta {∥}^{2}\;\mathrm{subject}\;\mathrm{to}∥\alpha {∥}^{2}=1$$
for any  λ> 0. Then, $sPC_{1}$ is derived as $sPC_{1}=x_{k}^{\left(j\right)}\frac{\hat{\beta }}{∥\hat{\beta }\;∥}$, where $x_{i}\;$denotes the ith row vector of the matrix $X_{k}^{\left(j\right)}$ [10]. Among the generated series of sPCs, we used the first sPC as a representative variable.

We then apply this procedure to every block, in each omics dataset. Once the variable is divided into $K_{j}$ blocks, we can then construct a new dataset $Y^{\left(j\right)}\;=\;\left[sPC_{1}^{\left(j\right)}\left|\cdots \right|sPC_{K_{j}}^{\left(j\right)}\right]$ for the jth omics dataset, where $\;sPC_{k}^{\left(j\right)}$ represents the first sPC for the kth block of the jth omics data set$\;\left(k=1,\;\cdots ,K_{j},\;\;j=1,\;\cdots ,\;J,\;\;K_{j}\ll p_{j}\right)$.

### 4.2. Stage 2: Integrative Analysis for Multi-Omics Data

In this stage, we firstly integrate the information from the omics datasets in parallel, and then construct a prediction model using the sPCs obtained from stage 1. We also assess the performance of the model, using AUC, and visualize the results by a graph.

#### 4.2.1. Parallel Integration of Omics Data, and Construction of a Prediction Model

Here, we combine the sPCs, from every omics data set and construct a new dataset $\;Y=\left[Y^{1}\left|Y^{2}\right|\cdots |Y^{\left(J\right)}\right]$. Then, the new dataset has  n rows and $K^{*}=\sum_{j=1}^{J}K_{j}$ columns. For example, in the GBM data analysis, the new dataset consisted of 215 rows of patients and 32 columns of sPCs from three omics datasets. Set an  n×2 matrix, Z, consisting of two variables of survival time T, and a censoring indicator, D. We can then consider Cox regression with the hazard function as:(6)$$\;h\left(t\right)=h_{0}\left(t\right)\times exp\left(b_{1}sPC_{1}^{\left(1\right)}+\cdots +b_{K^{*}}sPC_{K}^{\left(J\right)}\right)$$
where $h_{0}\left(t\right)$ is a baseline hazard function, and a stepwise selection procedure is applied to select variables. Among various stepwise procedures, we used Akaike information criterion (AIC) in this study [66].

#### 4.2.2. Assessing Prediction Performance

To evaluate predictive power, we use evaluation measures such as the sensitivity and specificity for time-to-event data [67]. We then divide the observations into two classes, above- and below-median survival times. Then, we evaluate whether our model classifies individuals properly. This classification was evaluated using cumulative/dynamic, time-dependent receiver operator characteristic (ROC) curves, corresponding to AUC values [68]. The sensitivity and specificity are then defined, at each time point, and are denoted as cumulative sensitivity and dynamic specificity. Given the cumulative sensitivity and dynamic specificity, the cumulative/dynamic ROC curve was plotted with 1-specificity, at the x-axis, and sensitivity at the y-axis, as in an ordinary ROC curve. The cumulative/dynamic ROC curve, and corresponding AUC value, can be derived using the cdROC() function of the R package nsROC.

Let Q denote the diagnostic marker and  T denote the survival time. Then, the cumulative sensitivity $Se^{C}$ and dynamic specificity $Sp^{D}$ are defined as follows [68].
(7)$$Se^{C}\left(q,t\right)=P\left(Q>q|T\leq t\right)$$
(8)$$Sp^{D}\left(q,t\right)=P\left(Q\leq q|T>t\right)$$

Given the cumulative sensitivity and dynamic specificity, the cumulative/dynamic ROC curve was plotted with 1-specificity, at the x-axis, and sensitivity at the y-axis, as in an ordinary ROC curve. The cumulative/dynamic ROC curve, and corresponding AUC value, can be derived using the cdROC() function of the R package nsROC.

We also compute Harrell’s C-index, also known as concordance index, which has been commonly used to assess and compare the discriminative power of risk prediction models [69]. It can be expressed in a formula
(9)$$C=\frac{\sum_{i\neq j}I\left(\eta_{i}<\eta_{j}\right)I\left(T_{i}>T_{j}\right)d_{j}}{\sum_{i\neq j}I\left(T_{i}>T_{j}\right)d_{j}}$$
where $\eta_{i}$ and $T_{i}$ are risk score and time to event for the ith patient, respectively; $d_{i}\;$denotes event indicator for the ith patient, with 0 for censoring and 1 for observed event; and$\;I\left(\cdot \right)$ is an indicator function [70]. The C-index can be interpreted as the fraction of all pairs of subjects whose predicted survival times are correctly ordered among all subjects who can actually be ordered [71]. We used the coxph() function of the R package ’Survival’ to calculate the C-index.

#### 4.2.3. Visualization

For visualization of the results, from the two stages together, we propose a new graph representing the estimated coefficients of selected sPCs, and loadings of the original variables, consisting of sPCs. Since this shows two levels of coefficients, from the Cox regression model and sPCA analysis, in a polar chart, we call it a multi-level polar chart (MP chart).

Suppose that $K^{*}\;$sPCs are selected from stage 2. Draw a circle and divide the angle equally into $K^{*\;}$pieces. For the first level chart, sort sPCs by the absolute value of the coefficients in the Cox regression model, and determine the corresponding sector clockwise. Adjust the radius of each sector to be proportional to the absolute value of the coefficient of the corresponding sPCs, $\left|b_{k}\right|$. The signs of coefficients are distinguished by the color of each sector. For example, the white sector indicates that the coefficient of the corresponding sPC is negative. At the second level of the chart, to represent how the sPCs are constructed, we plot the associated original variables with sPCs in the sector. Then, the distance from the origin of each variable is proportional to the corresponding loading of sPC. The sign is represented by the color and/or shape of the point. For example, white circles in a sector indicate that the variable loading is negative, while black triangles indicate that the variable loading is positive. Jittering would be done to avoid duplicates, and to clearly distinguish between variable points. Thus, the MP chart can simultaneously represent the important sPCs for survival time prediction, and what variables make up those sPCs. An illustrative example of our plots is presented in Figure 6.

## Figures and Tables

**Figure 1 ijms-21-08202-f001:**
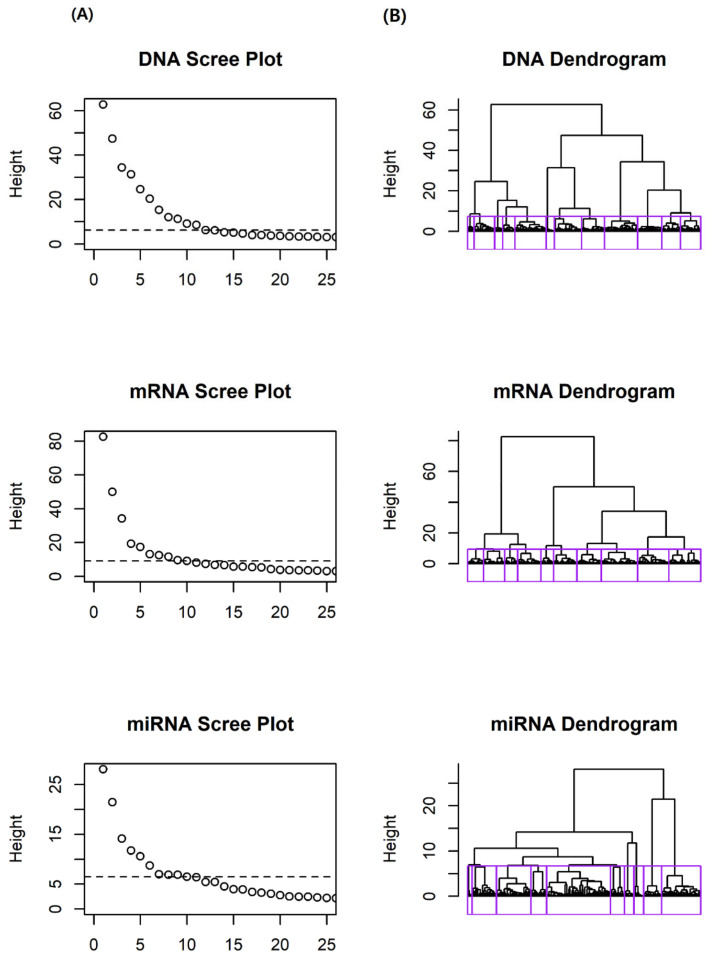
Scree plots and dendrograms for GBM data. (**A**) Scree plot of dissimilarity according to number of clusters. Line in the scree plot represent cut-off lines. (**B**) Dendrogram from hierarchical clustering. Rectangles in dendrogram represent homogeneous variable blocks.

**Figure 2 ijms-21-08202-f002:**
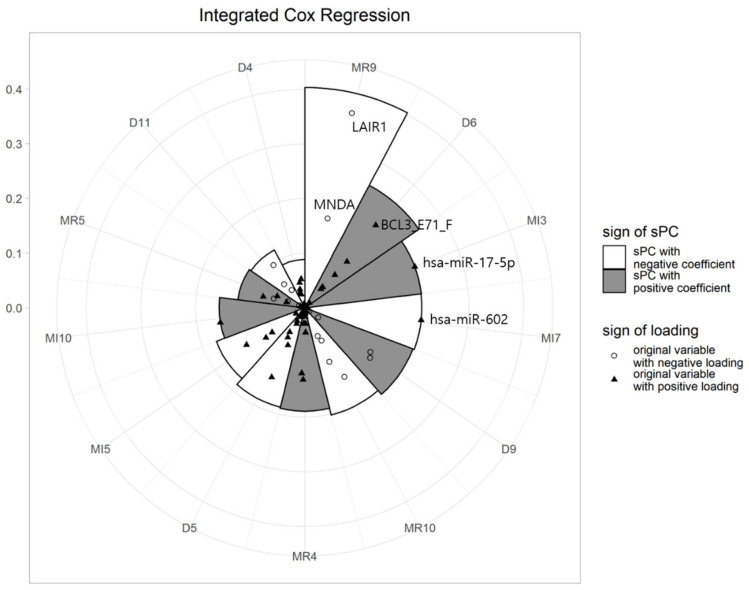
MP chart of integrated dataset analysis (MR: mRNA, D:DNA methylation, MI:miRNA).

**Figure 3 ijms-21-08202-f003:**
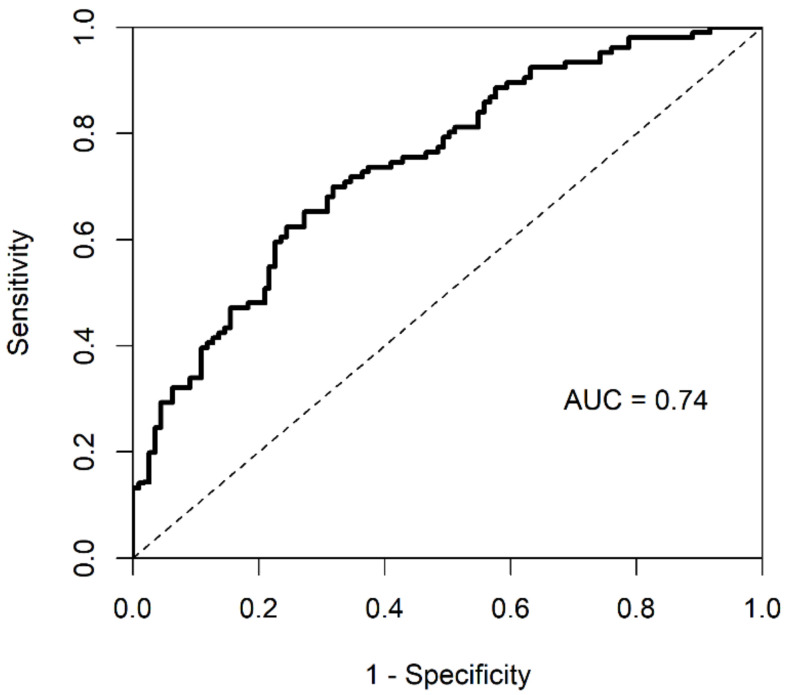
ROC curve and AUC value of integrated dataset by iMO-BSPC.

**Figure 4 ijms-21-08202-f004:**
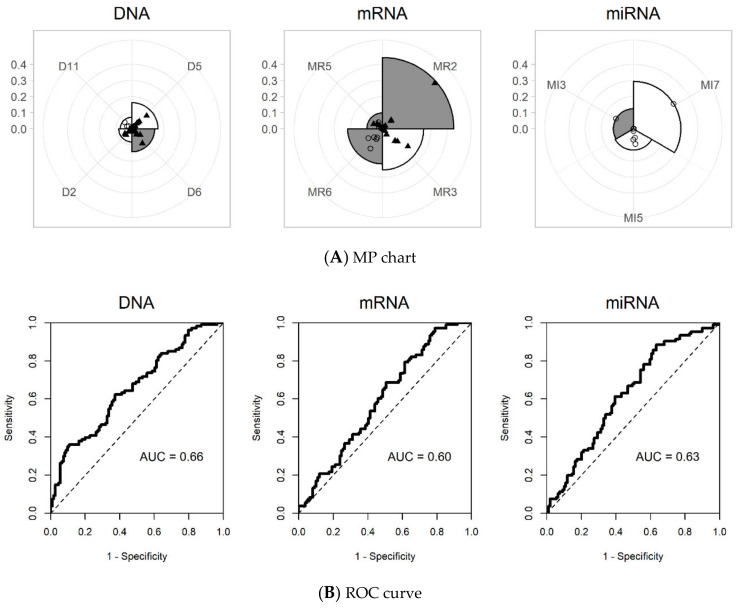
MP charts and ROC curves for each single omics dataset using iMO-BSPC. (**A**) MP chart. (**B**) ROC curve (DNA: DNA methylation, mRNA: mRNA expression, miRNA: miRNA expression).

**Figure 5 ijms-21-08202-f005:**
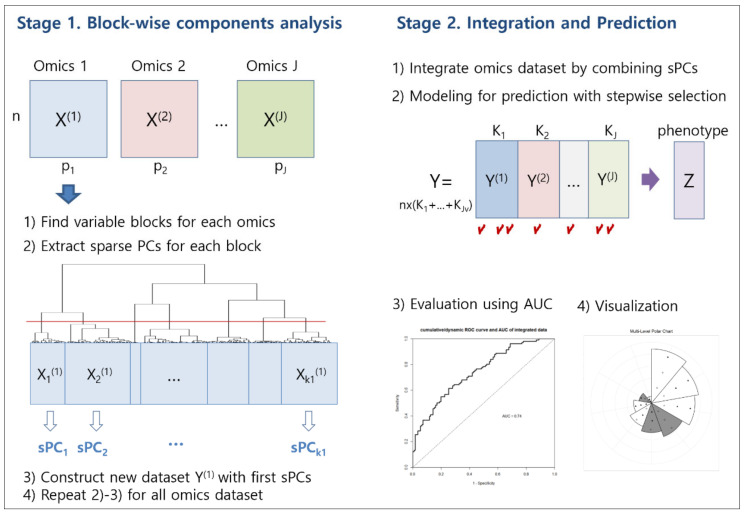
Procedure of iMO-BSPC.

**Figure 6 ijms-21-08202-f006:**
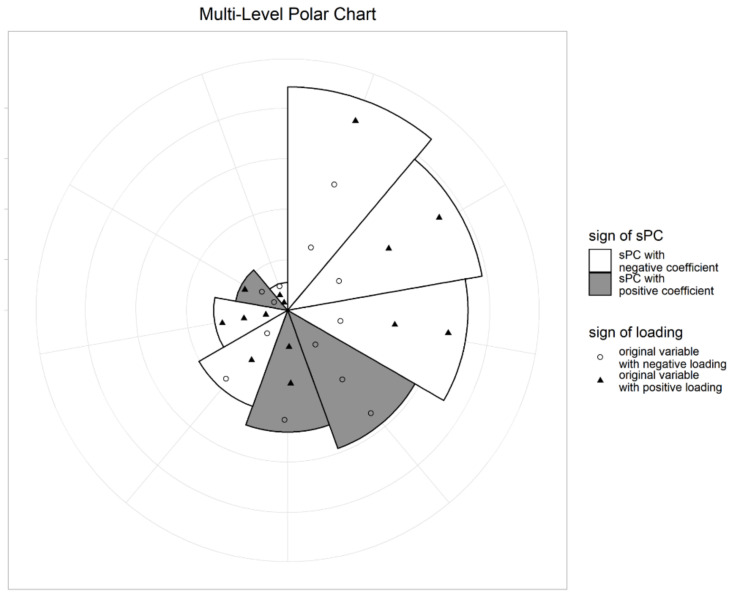
Example of a multi-level polar chart (MP chart): the radius of a sector is proportional to the coefficient of the sPC, and the distance from the origin to specific points is proportional to the variable loading.

**Table 1 ijms-21-08202-t001:** Stepwise Cox regression results for the integrated datasets.

Omics	Block	Number of Variables	Number of Variables Remained	Coefficient (Standard Error)	Hazard Ratio	^1^*p*-Value
DNA	D4	128	13	−0.09 (0.05)	0.92	0.089
	D5	186	19	−0.19 (0.05)	0.83	<0.001
	D6	153	15	0.25 (0.06)	1.29	<0.001
	D9	69	7	0.21 (0.07)	1.24	0.004
	D11	46	5	−0.12 (0.05)	0.89	0.011
mRNA	MR4	188	19	0.19 (0.07)	1.21	0.009
	MR5	161	16	0.12 (0.06)	1.13	0.033
	MR9	65	6	−0.40 (0.10)	0.67	<0.001
	MR10	67	7	−0.20 (0.07)	0.82	0.011
miRNA	MI3	32	3	0.21 (0.08)	1.24	0.005
	MI5	79	8	−0.17 (0.05)	0.84	<0.001
	MI7	56	6	−0.21 (0.09)	0.81	0.021
	MI10	22	2	0.16 (0.06)	1.17	0.016

^1^ Uncorrected *p*-value.

**Table 2 ijms-21-08202-t002:** Comparisons of predictability with other approaches.

Methodology	Single Omics									Multi-Omics		
	(1)			(2)			(3)			(1)	(2)	(3)
Omics	(a)	(b)	(c)	(a)	(b)	(c)	(a)	(b)	(c)	all	all	all
^1^ PC-before	12	10	10	12	10	10	12	10	10	32	32	32
^2^ PC-after	5	2	2	8	4	3	4	4	3	13	15	13
^3^ variable	563	188	225	935	559	167	56	49	17	1580	1339	126
AUC	0.67	0.54	0.61	0.71	0.58	0.59	0.66	0.60	0.63	0.75	0.76	0.74
C-index	0.63	0.55	0.60	0.64	0.61	0.60	0.61	0.59	0.60	0.69	0.69	0.67

(1) PCA without variable clustering, (2) PCA with clustering, (3) iMO-BSPC; (a) DNA methylation, (b) mRNA expression, (c) miRNA expression; ^1^ Number of PCs (sPCs) before stepwise variable selection process; ^2^ Number of PCs (sPCs) selected by stepwise variable selection process; ^3^ Number of variables used for prediction.

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
