# Peer review of "Integrative Analysis of Multi-Omics Data Based on Blockwise Sparse Principal Components"

_ijms, 2020, doi:10.3390/ijms21218202_

Round 1

Reviewer 1 Report

The authors propose a new methodological approach to the processing of multi-omics data. I consider the idea of using dimension reduction and subsequent integration to be an excellent scientific contribution. I give a major revision with regard to the fact that the methodology is not adequately substantiated in the manuscript. The discussion completely lacks a comparison with the current literature and there is not supported the robustness of the proposed methodology.

Below I provide comments that should be mentioned to understand the proposed methodology and its results. The notes are divided into 2 groups (major and minor) according to their importance.

Major notes

  1. The difference between method and methodology is not clear enough in the article. The authors present the "proposed method iMO-BSPC", which, according to the description, is more reminiscent of a methodology. It is containing previously created methods and data analysis tools (such as sPCA, cluster algorithms, etc.).
  2. Line 168: "...using only 10% of the variables, from these two methods." It is not clear if component analysis extracts really 10% of variables or 10 % of variance as is standard.
  3. All images are generally of very poor quality. Some appear to have been created with the surface snippet tool (eg Figure 5, where the underlined text is probably from the text editor where the image was created). If the authors state as one of their results the graphical tool for visualization of results ("We also propose a graphical method showing the results of sparse PCA ...."), then the images should be of adequate quality.
  4. Discussion: There is no comparison of whether someone did a similar analysis on a similar dataset with any of the mentioned approaches, or what is generally considered as a good approach in the literature.
  5. In general, both PCA and sPCA have linear relationships, haven't you considered a nonlinear approach? If not, why not?
  6. In general, the discussion is very short, there is no comparison with any literature - references. If the authors did not compare their results with other values, how do they know that their algorithm is better? No limitations of the methodology are discussed, is an ideal case without limitations proposed?
  7. The chosen 10% threshold is not discussed (comparison with other threshold values or methodology of ideal threshold choose).

Minor notes

  1. The introduction is very closely focused on the topic, which sometimes leads to ambiguity. For example, line 55: "However, this type of variable selection can duplicate highly correlated variables, ..." is not supported by the literature, so the reader cannot determine in which cases it may occur.
  2. Introduction: For better clarity, I suggest to pay more attention to the current state than to the description of the methodology itself (lines 76 - 101).
  3. The text lacks cross-references to figures, which makes it difficult to orient in the manuscript.
  4. The text lacks cross-references to the tables too.
  5. 127 lines: it would be good to state in the text that the samples correspond to patients.
  6. Table 1: The power of E-XX jumped to the second row (I recommend unifying for all columns)
  7. Table 2: It is not clear if are the p-values uncorrected or corrected.
  8. The pictures generally have a very brief description and even in the text, it is not entirely clear everything that a person can observe from the pictures (eg shapes, in the case of multiple graphs in one picture, I recommend describing A, B, etc.).
  9. The first sentences of the discussion only repeat the information in the introduction about the created methodology, it is a redundant part of the discussion.
  10. Line 192: The claim “BSPC has the advantages of reducing variable redundancy, and increasing interpretability“ needs to be substantiated.
  11. The “blocks” are the same as “clusters”? If so, it is necessary to unify the terminology.
  12. Sometimes 3 parts are described, sometimes 2 parts of the methodology - it would be good to unify the number of parts.
  13. The equation in Figure 5 is worse to understand with respect to naming the variables in the same figure.
  14. Does the ClustOfVar algorithm have the number of clusters as the only setting - the input parameter? In the literature, ClustOfVar is mentioned as a package. It would be good to write the necessary requirements, SW version.
  15. I appreciate that the authors put their work and presented the procedure of partial steps of the algorithm using equations. However, sometimes the orientation is worse in relation to the concrete problem. The equations are described in general. I suggest that the authors try to describe partial matrices, vectors, and other variables more in relation to a particular dataset and thus facilitate the work of the reader.
  16. Line 268: There is a special bounce in the variable font.
  17. Section Abbreviations does not contain all used abbreviations in the manuscript (missing AUC, ROC, SAS software, etc.).

Reviewer 2 Report

There are several typing errors or format problems in Pages 11 and 12.  For example,

In Figure 5, $a_{2 q_2}$.

L228: The last P should be capitalized.

L246: Need a period after “blocks”.

L253, L254: There is some problem in display the math formula.

L254, L255, L256: The word break seems to have some problem.

L267: an

L314, 315, 316: word break

Author Response

Our point-by-point responses

There are several typing errors or format problems in Pages 11 and 12.  For example,

In Figure 5, $a_{2 q_2}$.

[Answer] It's been corrected.

L228: The last P should be capitalized.

[Answer] It's been corrected.

L246: Need a period after “blocks”.

[Answer] It's been corrected.

L253, L254: There is some problem in display the math formula.

[Answer] It's been corrected.

L254, L255, L256: The word break seems to have some problem.

[Answer] It's been corrected.

L267: an

[Answer] It's been corrected.

L314, 315, 316: word break

[Answer] It's been corrected.

Reviewer 3 Report

One big concern about this work is the absence of rigorous comparison with existing methods of omics data analysis and integration. To the least, could the authors process the datasets through a simple PCA analysis and show the results? 

Move Table 1 to Supplementary.

Increase the font size in Figure 4.

Once the clusters are agglomerated, how do the authors decide on the final set of clusters? what filtering/threshold parameters are used in this step?

Reviewer 4 Report

In this paper, the authors have proposed a novel integrative method to analysis multi-omics data.  The method is well motivated and explained, and the paper is also well structured.  The presentation of the formulas and the algorithm is clear. However, several issues should be addressed before publication.

1. Splitting the samples patient's survival time  above or below the median survival time, and calculate the AUC value to evaluate the integrative perormance might be a problem.  This threshold is somewhat arbitrary. There are metrics like Brier, c-index, and others that enable the evaluation of prognostic accuracy without setting an arbitrary threshold.

2. It's nice that the models were interpreted, but this doesn't really provide any important insights from the biological side. 

3. There are quite some typographical and grammatical errors which need to be rectified before publication.

Round 2

Reviewer 1 Report

Dear authors, first of all, I would like to appreciate your efforts and the incorporation of all comments. The manuscript is easier to follow due to partial methodological procedures in the current version. There is still an opportunity to substantiate the robustness of the results, however, even in this way the description of the results, including the discussion, has been significantly modified. In the future, I would suggest further testing your methodology and substantiating its benefits. With regard to the made modifications, I propose to accept the article for publication.

Reviewer 4 Report

The authors have addressed all my comments